# FEDERATED LEARNING WITH LOCAL OPENSET NOISY LABELS

## ABSTRACT

Federated learning is a learning paradigm that allows the central server to learn from different data sources while keeping the data private locally. Without controlling and monitoring the local data collection process, the locally available training labels are likely noisy, *i.e.*, the collected training labels differ from the unobservable ground truth. Additionally, in heterogenous FL, each local client may only have access to a subset of label space (referred to as openset label learning), meanwhile without overlapping with others. In this work, we study the challenge of federated learning with local openset noisy labels. We observe that many existing solutions in the noisy label literature, *e.g.*, loss correction, are ineffective during local training due to overfitting to noisy labels and being not generalizable to openset labels. To address the problems, we design a label communication mechanism that shares randomly selected "contrastive labels" among clients. The privacy of the shared contrastive labels is protected by label differential privacy (DP). Both the DP guarantee and the effectiveness of our approach are theoretically guaranteed. Compared with several baseline methods, our solution shows its efficiency in several public benchmarks and real-world datasets under different noise ratios and noise models.

## 1 INTRODUCTION

Data heterogeneity is a common issue among different data centers. The label spaces of the data centers are likely different due to the heterogeneity of data sources. For example, the virus variants during the pandemic may differ in different regions, leading to an extremely heterogeneous data distribution among data centers. The heterogeneity challenges collaborations among data centers, *e.g.*, federated learning (FL), where each data center joins as a client to train a uniform and stronger global model for all the regions without sharing the sensitive data. In addition to a heterogeneous label space, what makes matters worse is that the observed label space may be noisy due to the limited knowledge access between different data centers, making this problem more challenging. This paper aims to provide solutions for a practical FL setting where not only do each client's training labels carry different noise rates, but the observed label space at these clients can also be noisy and differ, even though their underlying clean labels are drawn from the same label space. We call that such a federated learning system has *local openset noise* problems if the observed label space is noisy and differs across clients.

The above local openset label noise will pose significant challenges if we apply the existing learning with noisy label solutions locally at each client. For instance, a good number of these existing solutions operate with centralized training data and rely on the design of robust loss functions (Natarajan et al., 2013; Patrini et al., 2017; Ghosh et al., 2017; Zhang & Sabuncu, 2018; Feng et al., 2021; Wei & Liu, 2021; Zhu et al., 2021a). Implementing these approaches often requires assumptions, which are likely to be violated if we directly employ these centralized solutions in a federated learning setting. For example, loss correction is a popular design of robust loss functions (Patrini et al., 2017; Natarajan et al., 2013; Liu & Tao, 2015; Scott, 2015; Jiang et al., 2022), where the key step is to estimate the label noise transition matrix correctly (Bae et al., 2022; Zhang et al., 2021b; Zhu et al., 2021b; 2022). Correctly estimating the label noise transition matrix requires observing the full label space, when the ground-truth labels are unavailable. In FL, where the transition matrix is often estimated only with the local openset noisy labels, existing estimators of the noise transition matrix would fail. Moreover, even though we can have the best estimate of the noise transition matrix if we

have the ground-truth labels for the local instances, the missing of some label classes would make the estimate different from the ground-truth one, and again leads to failures (detailed example in Section 3.1).

Intuitively, we may share some label information among the clients to generalize some centralized training methods to FL. However, it is against privacy protection, making it challenging in real usage. Moreover, it is also important to figure out what kind of label information is sufficient to solve the local openset noisy problems in FL. In this paper, we use the global label distribution as a hint to local clients, where the hint is used in a contrastive way to avoid overfitting to noisy labels. To protect privacy during label communication, we randomly flip the shared labels to ensure label differential privacy (DP). Our contributions are summarized as follows.

- We formally define the openset noise problem in FL, which is more practical than the existing heterogeneous noisy label assumptions. The challenges along with the openset noise are also motivated by analyzing the failure cases of the existing popular noisy learning solutions such as loss correction (Natarajan et al., 2013; Patrini et al., 2017; Liu & Tao, 2015).
- We propose a novel framework, FedDPCont, to solve the openset label noise problem, which builds on the idea of using globally shared private contrastive labels to avoid overfitting to local noisy labels.
- To mitigate the gap between the centralized usage of noisy labels and the federated one, we propose a *label communication* algorithm with a differential privacy (DP) guarantee. We also prove that benefiting from label communication, the gradient update of aggregating local loss with private labels is guaranteed to be the same as the corresponding centralized loss, and further establish its robustness to label noise.
- We empirically compare FedDPCont with several baseline methods on both benchmark datasets and practical scenarios, showing that, in terms of FL with openset label noise, directly applying centralized solutions locally cannot work and FedDPCont significantly improves the performance.

## 2 RELATED WORKS

Federated learning is a collaborative training method to make full use of data from every client without sharing the data. FedSGD (Shokri & Shmatikov, 2015) is the way of FL to pass the gradient between the server and the clients. To improve the performance, FedAvg (McMahan et al., 2017) is proposed and the model weight is passed between the server and the clients. In practice, openset problem is common in FL because the source of every client may vary a lot and it is very likely to find that some of the classes are unique in the specific clients. There are a lot of works to analyze and solve the non-IID problem in FL (Zhao et al., 2018; Li et al., 2019; 2021; Zhang et al., 2021a; Li et al., 2020b; Karimireddy et al., 2020; Andreux et al., 2020).

Label noise is common in the real world (Agarwal et al., 2016; Xiao et al., 2015; Zhang et al., 2017; Wei et al., 2022b). Traditional works on noisy labels usually assume the label noise is *class-dependent*, where the noise transition probability from a clean class to a noisy class *only* depends on the label class. There are many statistically guaranteed solutions based on this assumption (Natarajan et al., 2013; Menon et al., 2015; Liu & Tao, 2015; Liu & Guo, 2020). However, this assumption fails to model the situation where different group of data has different noise patterns (Wang et al., 2021). For example, different clients are likely to have different noisy label spaces, resulting totally different underlying noise transitions. Existing works on federated learning with noisy labels mainly assume the noisy label spaces are identical across different clients (Yang et al., 2022; Xu et al., 2022). There are other notable centralized solutions relying on the memorization effect of a large model (e.g., deep neural network) (Li et al., 2020a; Liu, 2021; Song et al., 2019; Xia et al., 2021; Liu et al., 2020; Cheng et al., 2020). However, in a federated learning system, simply relying on the memorization effect would fail, i.e., the model can perfectly memorize all local noisy samples during local training, since the local data is likely to be imbalanced and with a limited amount (Han et al., 2020; Liu, 2021). The idea of contrastive labels is to punish the overfitting, which is supposed to avoid memorizing openset local noisy samples. Besides, the concept "openset" is also used in Tuor et al. (2021), where the focus is on the out-of-distribution features and their labels are called openset noise. It is different from ours since they did not focus on in-distribution mislabeled data.

## 3 FORMULATIONS AND MOTIVATIONS

**Federated learning**    Consider a $K$ class classification problem in a federated learning system with $C$ clients. Each client $c \in [C] := \{1, \cdots, C\}$ holds a local dataset $D_c := \{(x_n^c, y_n^c)\}_{n \in [N_c]}$, where $N_c$ is the number of instances in $D_c$ and $N_c := \{1, \cdots, N_c\}$. Assume there is no overlap among $D_c, \forall c$. Denote the union of all the local datasets by $D := \{(x_n, y_n)\}_{n \in [N]}$. Clearly, we have $D = \cup_{c \in [C]} D_c$ and $N = \sum_{c \in [C]} N_c$. Denote by $\mathcal{D}_c$ the local data distribution, $(X^c, Y^c) \sim \mathcal{D}_c$ the local random variables of feature and label, $\mathcal{D}$ the global/centralized data distribution, and $(X, Y) \sim \mathcal{D}$ the corresponding global random variables. Denote by $\mathcal{X}$, $\mathcal{X}_c$, $\mathcal{Y}$, and $\mathcal{Y}_c$ the space of $X$, $X_c$, $Y$, and $Y_c$, respectively. FL builds on the following distributed optimization problem: $\arg\min_\theta \quad \sum_{c \in [C]} \frac{N_c}{N} \cdot L_c(\theta)$, where $\boldsymbol{f}$ is the classifier, $\theta$ is the parameter of $\boldsymbol{f}$. $\boldsymbol{f}$ and $f$ stand for the same model but different output. $f := \arg\max_{i \in [K]} \boldsymbol{f}$. To this end, the local training and global model average are executed iteratively. In local training, each client learns a model $\boldsymbol{f}_c : \mathcal{X} \to \mathcal{Y}$ with its local dataset $D_c$ by minimizing the empirical loss $L_c(\theta_c)$ defined as: $L_c(\theta_c) := \frac{1}{N_c} \sum_{n \in [N_c]} \ell(\boldsymbol{f}_c(x_n^c; \theta_c), y_n^c)$, where for classification problems, the loss function is usually the cross-entropy (CE) loss: $\ell(\boldsymbol{f}(X; \theta), Y) = -\ln(f_{(X;\theta)}[Y]), Y \in [K]$, indicating taking the negative logarithm of the $Y$-th element of $\boldsymbol{f}$ given input $X$ and model parameter $\theta$. In the following global model average, each client $c$ sends its model parameter $\theta_c$ to the central server, which is further aggregated following FedAvg (McMahan et al., 2017): $\theta = \sum_{c \in [C]} \frac{N_c}{N} \cdot \theta_c$.

### 3.1 OPENSET NOISE IN FEDERATED LEARNING

When the label $y$ is corrupted, the clean dataset $D$ becomes the noisy dataset $\tilde{D} := \{(x_n, \tilde{y}_n\}_{n \in [N]}$ where $\tilde{y}_n$ is the noisy label and possibly different from $y_n$. The noisy data $(x_n, \tilde{y}_n)$ can be viewed as the specific point of the random variables $(X, \tilde{Y})$ which is from the distribution $\tilde{\mathcal{D}}$. Noise transition matrix $T$ characterizes the relationship between $(X, Y)$ and $(X, \tilde{Y})$. The shape of $T$ is $K \times K$ where $K$ is the number of classes in $\mathcal{D}$. The $(i, j)$-th element of $T$ represents the probability of flipping a clean label $Y = i$ to noisy label $\tilde{Y} = j$, i.e., $T_{ij} := \mathbb{P}(\tilde{Y} = j | Y = i)$. If $\tilde{Y} = Y$ always holds, $T$ is an identity matrix. Note the above definition builds on the assumption that $T$ is class-dependent, which is a common assumption in centralized learning with noisy labels (Natarajan et al., 2013; Menon et al., 2015; Liu & Tao, 2015). However, in FL, $T$ is likely to be different for different clients (a.k.a. group-dependent (Wang et al., 2021)). Specifically, we use $T$ to denote the *global* noise transition matrix for $\tilde{D}$ and $T_c$ to denote the *local* noise transition matrix for $\tilde{D}_c$. In a practical federated learning scenario where the data across different clients are non-IID, different clients may have different label spaces. When the labels are noisy, we naturally have the following definition of openset label noise in FL.

**Definition 1** (Openset noisy labels in FL). *The label noise in client $c$ is called openset if $\tilde{\mathcal{Y}}_c \neq \tilde{\mathcal{Y}}$.*

**Generation of openset noise**    We propose the following noise generation process to model openset label noise in practical FL systems. Denote by $\mathbb{1}_{c,k}$ the indicator random variable that label class $k$ is included in client $c$, where $\mathbb{1}_{c,k} = 1$ (w.p. $Q_{c,k}$) indicates client $c$ has data belonging to class $k$ and $\mathbb{1}_{c,k} = 0$ otherwise. The indicators $\{\mathbb{1}_{c,k} \mid \forall c \in [C], k \in [K]\}$ are generated independently with the probability matrix $Q$, where the $(c, k)$-th element is $Q_{c,k} := \mathbb{E}[\mathbb{1}_{c,k}]$. In practice, if all the elements in $\{\mathbb{1}_{c,k} | k \in [K]\}$ are identical, meaning the client $c$ can observe nothing or all the classes, then $\{\mathbb{1}_{c,k} | k \in [K]\}$ will be re-generated until client $c$ is an openset client. Denote by $I_k := \{c | \mathbb{1}_{c,k} = 1, c \in [C]\}$ the set of clients that include class $k$. Denote by $\tilde{D}^{(k)} = \{n | \tilde{y}_n = k\}$ the indices of instances that are labeled as class $k$. For each $k \in [K]$, instances in $\tilde{D}^{(k)}$ will be distributed to clients with $\mathbb{1}_{c,k} = 1$ either uniformly or non-uniformly as follows.

- *Uniform allocation:* Randomly sample (without replacement) $|\tilde{D}^{(k)}|/|I_k|$ indices from $\tilde{D}^{(k)}$ and allocate the corresponding instances to client $c$. Repeat for all $c \in I_k$.
- *Non-uniform allocation:* Generate probabilities $\{u_c | c \in I_k\}$ from Dirichlet distribution $\mathsf{Dir}(\mathbf{1})$ with parameter $\mathbf{1} := [1, \cdots, 1]$ ($|I_k|$ values). Randomly sample (without replacement) $|\tilde{D}^{(k)}| \cdot u_c$ indices from $\tilde{D}^{(k)}$ and allocate the corresponding instances to client $c$. Repeat for all $c \in I_k$.

In this way, all the clients have openset label noise, i.e., $\mathcal{Y}_c \neq \tilde{\mathcal{Y}}, \forall c \in [C]$.

**Example**  Consider the following example. For a data distribution $(X, Y) \sim \mathcal{D}$ where $Y \in \mathcal{Y} :=$ $\{1, 2, \cdots, K\}$, the set of all the opensets is the combination of $\mathcal{Y}$ except the full set of $\mathcal{Y}$ and the empty set. For example, if $\mathcal{Y}$ is $\{1, 2, 3\}$, there would be $2^K - 2 = 6$ different combinations of the noisy label space: $\{1, 2, 3, (1, 2), (1, 3), (2, 3)\}$. It should be noted that it is still possible that the union of all the clients still cannot cover $\mathcal{Y}$. An example of the real and openset $T$ in the 3-class classification problem is as follows. Suppose the *real* noise transition matrix $T_{\text{real}}$ is shown on the LHS. However, if we only observe $\tilde{\mathcal{Y}}_c = \{1, 2\}$ in client $c$, the *optimal estimate* of $T$ relying only on $\tilde{D}_c$ could only be $T_{\text{OptEst}}$ even though we know $D_c$. This is because when $\tilde{\mathcal{Y}}_c = \{1, 2\}$, we have $\mathbb{P}(\tilde{Y} = 3) = 0 \Rightarrow \mathbb{P}(\tilde{Y} = 3|Y = 3) = 0$, resulting that the other two probabilities have to be normalized from $(1/16, 3/16)$ to $(1/4, 3/4)$ to get a total probability of 1.

$$T_{\text{real}} = \begin{bmatrix} 1 & 0 & 0 \\ 1/3 & 2/3 & 0 \\ 1/16 & 3/16 & 3/4 \end{bmatrix}, \ T_{\text{OptEst}} = \begin{bmatrix} 1 & 0 & 0 \\ 1/3 & 2/3 & 0 \\ 1/4 & 3/4 & 0 \end{bmatrix}$$

**Local openset noise is challenging**  A good number of correction approaches in the learning with noisy labels literature would require using the transition matrix $T$. For instance, loss correction (Patrini et al., 2017) is a popular tool to solve the noisy label problem as

$$\ell^{\rightarrow}(\boldsymbol{f}(X), \tilde{Y}) := \ell(T^{\top}\boldsymbol{f}(X), \tilde{Y}) \tag{1}$$

where $T^{\top}$ is the transpose of $T$. The key step of the loss correction approach is to estimate a correct $T$. However, if the label space is openset, the best estimated $T$ will lead to a wrong prediction result. Based on the example above, the best-corrected output is

$$T^{\top}\boldsymbol{f}(X) = \begin{bmatrix} 1 & 1/3 & 1/4 \\ 0 & 2/3 & 3/4 \\ 0 & 0 & 0 \end{bmatrix} \begin{bmatrix} f_1(X; \theta) \\ f_2(X; \theta) \\ f_3(X; \theta) \end{bmatrix} = \begin{bmatrix} f_1(X; \theta) + f_2(X; \theta)/3 + f_3(X; \theta)/4 \\ 2f_2(X; \theta)/3 + 3f_3(X; \theta)/4 \\ 0 \end{bmatrix}, \tag{2}$$

where $\boldsymbol{f} = [f_1, f_2, f_3]^{\top}$ and $f_n$ is the $n$-th element of $\boldsymbol{f}$. The model cannot distinguish class 3 which is reasonable. However, it will misclassify class 2 to class 3 because class 3 has a larger weight. For example, given an instance $(x, y = 2)$, the cross entropy loss is $-\ln(2f_2(x; \theta)/3 + 3f_3(x; \theta)/4)$ where $f_3(x; \theta) = 1$ leads to the minimization of the loss, making the loss correction fail.

## 3.2 Our Motivation and Building Idea

The above example highlights the challenge of adapting approaches that use noise transition matrix $T$ to our openset FL setting. Therefore, we hope to circle around by building our solutions upon ideas that do not require the knowledge of $T$.

According to the previous analyses, the main difficulty of local openset label noise exists in the mismatch of clean and noisy label spaces within a local client. Changing the label space is challenging in FL since it often requires sharing data between clients. Therefore, we need to solve two technical challenges here: 1) What kind of information can be shared to mitigate the heterogeneity introduced by local openset label noise? 2) How do we use the shared information to help training?

For the first challenge, we consider sharing the "private labels" since only sharing the label without disclosing features is usually less sensitive than sharing features in many cases, *e.g.*, face recognition. Additionally, it is relatively easier to protect the label privacy by random responses (Ghazi et al., 2021). For the second challenge, given only the private labels, we propose to use them "contrastively" to punish the overfitting of noisy labels. Intuitively, for a multi-class classification task, *e.g.*, 10 classes, a randomly picked private label $\check{y}_n$ is likely to be a wrong label for a randomly picked feature $x_{n'}$. Therefore, rather than guiding the model the memorize this pattern, we can just use it contrastively or negatively, i.e., $-\ell(\boldsymbol{f}(x_{n'}), \check{y}_n)$. Therefore, the new loss function with Private Labels becomes

$$\ell_{\text{PL}}(\boldsymbol{f}(x_n), \tilde{y}_n) := \ell(\boldsymbol{f}(x_n), \tilde{y}_n) - \ell(\boldsymbol{f}(x_n), \check{y}_{n'}). \tag{3}$$

The design is related to works such as (Wei et al., 2022a; Liu & Guo, 2020; Cheng et al., 2020), while the key difference is the selection of the labels for the second term, i.e., the private labels are drawn from the whole label space while directly using the above approach requires getting labels locally. Intuitively, the "new" label has to be sampled globally; otherwise, the global information is

missing and the negative effect of local openset label noise would induce performance degradation. Additionally, label communications in FL should be private. We defer the detailed explanation of its necessity to Appendix B.3.

## 4 PROPOSED METHOD

We propose the following label communication-aided algorithm FedDPCont, which we also illustrate in Figure 1. There are *two critical stages* to guarantee the success of the proposed methods with good DP protection:

- **Stage 1:** Privacy-preserving global label communication given in Section 4.1
- **Stage 2:** Contrastive gradient updates at the local client using $\ell_{\text{PL}}$ given in Section 4.2 and the shared label information from Stage 1.

### 4.1 LABEL COMMUNICATION

Label privacy protection is an essential feature of FL so we cannot pass $\tilde{Y}$ to the other clients, directly. To protect privacy, we adopt the label differential privacy (DP) as Definition 2.

**Definition 2** (Label Differential Privacy (Ghazi et al., 2021)). *Let $\epsilon > 0$. A randomized algorithm $\mathcal{A}$ is said to be $\epsilon$-label differentially private ($\epsilon$-labelDP) if for any two training datasets $D$ and $D'$ that differ in the label of a single example, and for any subset $S$ of outputs of $\mathcal{A}$,*

$$\mathbb{P}(\mathcal{A}(D) \in S) \leq e^{\epsilon} \cdot \mathbb{P}(\mathcal{A}(D') \in S).$$

The high-level idea is to achieve label privacy (DP), each client $c$ will use a symmetric noise transition matrix $T_{\text{DP}}$ to flip their local labels to protect their labelDP:

$$T_{\text{DP}}[y, \tilde{y}] := \mathbb{P}(\widetilde{Y} = \tilde{y}|Y = y) = \begin{cases} \frac{e^{\epsilon}}{e^{\epsilon}+K-1}, & \text{if } \tilde{y} = y, \\ \frac{1}{e^{\epsilon}+K-1}, & \text{if } \tilde{y} \neq y. \end{cases}$$

where $K$ is the number of classes. Then only the flipped labels are shared between the clients and the server. It is easy to show that sharing the flipped labels using $T_{\text{DP}}$ suffices to preserve labelDP:

**Theorem 1** (Label Privacy in FedDPCont). *Label sharing in FedDPCont is $\epsilon$-labelDP.*

Denote by $\tilde{\boldsymbol{p}}_n^c$ the one-hot encoding of $\tilde{y}_n^c$. The whole label communication process is presented in Algorithm 1. At the beginning of the algorithm, the server will initialize $T_{\text{DP}}$ according to $\epsilon$ and broadcast $T_{\text{DP}}$ to all $C$ clients. For each client $c$, it calculates the DP label distribution of every data point $(x_n^c, \tilde{y}_n^c)$ as $\check{\boldsymbol{p}}_n^c = T_{\text{DP}}^{\top}\tilde{\boldsymbol{p}}_n^c$, where $\check{\boldsymbol{p}}_n^c$ is the distribution of DP label in client $c$. With this distribution, the client generates the DP private label $\check{y}_n^c, n \in [N_c]$ for every data point and every client sends all $\check{y}_n^c$ back to the server. After obtaining all $\check{y}_n^c$ from the clients, the server aggregates the label and calculates the posterior label distribution $\check{\boldsymbol{p}}$. To restore the correct distribution of $\tilde{Y}$, the server calculates $(T_{\text{DP}}^{\top})^{-1}\check{\boldsymbol{p}}$. Note that

$$(T_{DP}^{\top})^{-1}T_{DP}^{\top}(\sum_{i=1}^{C} \tilde{\boldsymbol{p}}_n^c)/C = \tilde{\boldsymbol{p}}.$$

To apply $T_{\text{DP}}$ and $(T_{\text{DP}})^{-1}$ sequentially, FedDPCont enables the clients to share the information with the others while DP is guaranteed. Finally, the client calculates the local loss according to Equation (3) where $\tilde{Y}$ is sampled from $\mathbb{P}(\tilde{Y} = i) := ((T_{\text{DP}}^{\top})^{-1}\check{\boldsymbol{p}})[i]$. This label communication procedures guarantees $\epsilon$-DP.

### 4.2 FEDDPCONT

Based on the distribution $\mathbb{P}(\tilde{Y}|\tilde{D})$, we propose *FedDPCont*, a novel framework based on FedAvg, to solve the local openset noise problem. Denote by $\Delta_c^{(r)} := \theta_c^{r+1} - \theta_c^r$, the variation of model parameters in the $r$-th round of the local training in client $c$. Recall $\theta_c$ is the parameter of $\boldsymbol{f}_c$.

Denote by $\Delta^{(r)} := \theta^{r+1} - \theta^r$ the variation of model parameters in the $r$-th round of the corresponding global gradient descent update assuming the local data are collected to a central server. Define $\mathbb{P}(\mathcal{D}_c|\mathcal{D}) := \mathbb{P}((X, Y) \sim \mathcal{D}_c \mid (X, Y) \sim \mathcal{D})$. Numerically, it is calculated as $N_c/N$ for client $c$ given $D$. We have the following theorem for the calibration property of FedDPCont.

---

**Algorithm 1** Label Communication in FedDPCont

1: **Initialization:** The server initialize $T_{\text{DP}}$ according to $\epsilon$ and broadcast $T_{\text{DP}}$ to all clients.
   *# Client label differential privacy protection*
2: **for** $c$ in $C$ clients **do**
3:    calculate $\check{\boldsymbol{p}}_n^c = T_{\text{DP}}^\top \tilde{\boldsymbol{p}}_n^c, \forall n \in [N_c]$.
4:    generate the private label $\check{y}_n^c$ using $\mathbb{P}(\check{y}_n^c = i) = \check{\boldsymbol{p}}_n^c[i], \forall i \in [K], n \in [N_c]$.
5:    send $\{\check{y}_n^c\}_{n \in [N_c]}$ to the server
6: **end for**
7: The server aggregates the label $\{\check{y}_n^c\}_{n \in [N_c]}$ sent from all $C$ clients.
8: The server calculates the posterior label distribution $\check{\boldsymbol{p}}$: $\check{\boldsymbol{p}}[i] := \frac{1}{N} \sum_{c=1}^{C} \sum_{n=1}^{N} \mathbb{1}(\check{y}_n^c = i)$.
9: The server calculates $(T_{\text{DP}}^\top)^{-1} \check{\boldsymbol{p}}$ and sends it to each client $c$.
10: The client $c$ samples the $\tilde{y}_{n'}$ in Eqn. (3) following $\mathbb{P}(\tilde{Y} = \tilde{y}_{n'}) = ((T_{\text{DP}}^\top)^{-1} \check{\boldsymbol{p}})[\tilde{y}_{n'}]$.

---

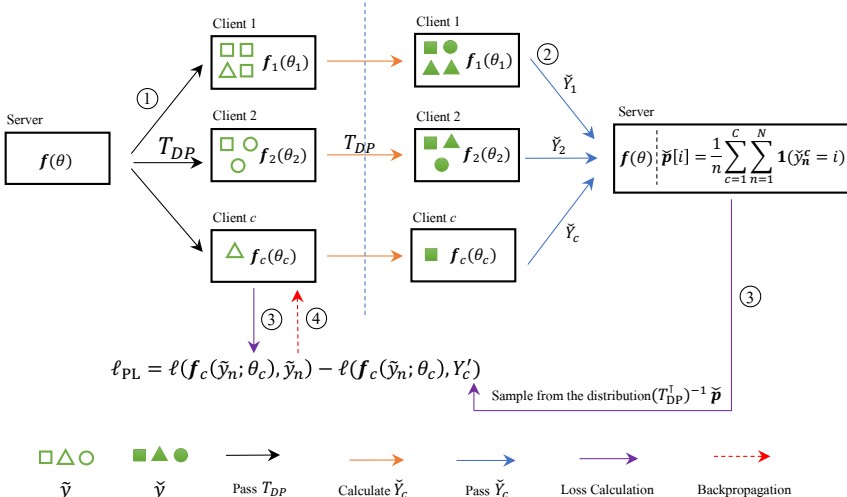

Figure 1: The illustration of FedDPCont. *Step 1* is the $T_{\text{DP}}$ generation where the server generates $T_{\text{DP}}$ according to $\epsilon$ and sends it to each client. After receiving $T_{\text{DP}}$, *Step 2* is the label communication. Every client $c$ calculates DP label $\check{Y}_c$ according to $T_{\text{DP}}$ and the noisy label $\tilde{Y}_c$. Clients send $\check{Y}_c$ to the server. The server aggregates every $\check{Y}_c$, calculates the posterior label distribution $\check{\boldsymbol{p}}$ and sends $(T_{\text{DP}}^\top)^{-1} \check{\boldsymbol{p}}$ to every client for the contrastive term sampling. *Step 3* is the loss calculation using the noisy label $\tilde{Y}_c$ on every client $c$, the model prediction $\hat{Y}_c$ and $Y_c'$ sampled from $(T_{\text{DP}}^\top)^{-1} \check{\boldsymbol{p}}$ and calculate loss. *Step 4* is the back-propagation for contrastive gradient updates.

**Theorem 2** (Local clients with FedAvg). *The aggregated model update of FedDPCont is the same as the corresponding centralized model update, i.e.,*

$$\sum_{c \in [C]} \mathbb{P}(\mathcal{D}_c | \mathcal{D}) \cdot \Delta_c^{(r)} = \Delta^{(r)},$$

Theorem 2 shows that the extra effect of local openset label noise can be mitigated by sharing private labels and FedAvg. Note the theorem only discusses the case in the expectation level (infinite data size), meaning the gap between distributed learning and centralized learning given limited data still exists. Given Theorem 2, we can further show $\ell_{\text{PL}}$ is robust to label noise as what has been done for centralized training (Liu & Guo, 2020).

The details of FedDPCont are shown in Algorithm 2. At the beginning of FedDPCont, the server and the clients $c$ initialize the model and each client $c$ initialize its own dataset $D_c = \{X_c, \tilde{Y}_c\}$ and loss function $\ell$. After this, the server generates the DP matrix $T_{\text{DP}}$ and sends it to every client and every client $c$ can generate DP labels $\check{y}_n^c$. Next, every client $c$ sends $\check{y}_n^c$ to the server, and the server aggregates DP labels according to Section 4.1. After aggregation at the server, a posterior label

---

**Algorithm 2** FedDPCont.

---

1: **Server:** initialize model $\boldsymbol{f}_g$, global step size $\alpha_g$ and global communication round $R$.
2: **Each Client** $c$**:** initialize model $\boldsymbol{f}_c$ , the dataset $D_c = \{(x_n^c, \tilde{y}_n^c)\}_{n \in [N_c]}$, local learning rate $\alpha_c$ and local updating iterations $E$.
3: The server generates and broadcasts $T_{\mathrm{DP}}$ to all clients according to Definition 2.
4: Clients generate DP labels $\breve{y}_n^c$ and send $\breve{y}_n^c$ to the server according to Section 4.1.
5: The server aggregates $\breve{y}_n^c$ and calculate the posterior label distribution $\breve{\boldsymbol{p}}$.
6: The server send $(T_{\mathrm{DP}}^\top)^{-1}\breve{\boldsymbol{p}}$ to each client.
7: **for** $i = 1 \to R$ **do**
8:     Randomly select $C'$ clients from $C$ according to $\lambda$
9:     **for** $c$ in $C'$ clients **do**
10:         $\boldsymbol{f}_c \leftarrow \boldsymbol{f}$
11:         **for** $j = 1 \to E$ **do**
12:             $\hat{y}_n^c \leftarrow \boldsymbol{f}_c(x_n^c), \forall n \in [N_c]$
13:             Sample $(y_c^n)'$ following $(T_{\mathrm{DP}}^\top)^{-1}\breve{\boldsymbol{p}}, \forall n \in [N_c]$.
14:             $L_c \leftarrow \frac{1}{N_c}\sum_{n=1}^{N_c}(\ell(\hat{y}_n^c, \tilde{y}_n^c) - \ell(\hat{y}_n^c, (y_c^n)'))$
15:             $\boldsymbol{f}_c \leftarrow \boldsymbol{f}_c - \alpha_c \cdot \nabla L_c$
16:         **end for**
17:     **end for**
18:     $\boldsymbol{f} \leftarrow \boldsymbol{f} - \alpha_g \cdot \sum_{c=1}^{C'}(\boldsymbol{f}_c - \boldsymbol{f})$
19: **end for**

---

distribution $\breve{\boldsymbol{p}}$ can be computed and the server sends $(T_{\mathrm{DP}}^\top)^{-1}\breve{\boldsymbol{p}}$ back to the client so that the client can sample the private label from this distribution. To simulate the practical usage, only part of rather than all clients will participate in the training in one round. The clients are chosen randomly according to the federated fraction $\lambda$. The selected clients sample the "contrastive label" $Y_c'$ from the distribution $(T_{\mathrm{DP}}^\top)^{-1}\breve{\boldsymbol{p}}$ and calculate the loss $L_c$ according to Eq. (3) by using the output of the model $\hat{Y}_c$. The model weight is updated by $L_c$ and the server weight is averaged according to FedAvg (McMahan et al., 2017), which is the end of one communication round.

**Privacy Issue.** We are aware that the label distribution recovered by our algorithm may also be a concern of privacy. However, the existing works about the attack in federated learning are mainly from embedding layers Melis et al. (2019), fully-connected layers Zhao et al. (2020); Geiping et al. (2020); Pan et al. (2020), and model gradients Aono et al. (2017); Melis et al. (2019). Different from the leakage of individual labels, the recovered label distribution by our algorithm has much less information. There is no direct evidence of the harm of leaking an imperfect label distribution to the best of our knowledge. In Table 3, we will illustrate that different DP privacy level ($\epsilon$) corresponds to different performance, indicating that, even though we have restored the distribution of $\widetilde{Y}$ (Algorithm 1, Line 9), it is still different from the original one.

## 5 EXPERIMENTS AND RESULTS

### 5.1 EXPERIMENTS SETUP

To validate the generality and effectiveness of FedDPCont, we select several public datasets with various levels of difficulties, including CIFAR-10, CIFAR-100 (Krizhevsky et al., 2009) as benchmark datasets and CIFAR-N (Wei et al., 2022b), Clothing-1M (Xiao et al., 2015) as real-world datasets. To simulate the practical usage, we first apply the noise on the label and generate the openset candidates according to the number of classes $K$ for every client because only the noisy label is visible to the client in the real world. On CIFAR-10 and CIFAR-100, we apply the symmetric noise for benchmark testing while we apply random noise for practical simulation. Furthermore, we also test the performance using Clothing-1M and CIFAR-N to test the performance of FedDPCont in real-world scenarios.

For baseline methods, we use FedAvg (McMahan et al., 2017), forward loss correction (LC) (Patrini et al., 2017), FedProx (Li et al., 2020b), Co-teaching (Han et al., 2018) and T-revision (Xia et al., 2019), FedBN (Li et al., 2021), FedDyn (Acar et al., 2021), Scaffold (Karimireddy et al., 2020)

Table 1: The performance (the best accuracy) of all methods on CIFAR-10 and CIFAR-100. FedDPCont is always the best method.

| Dataset | Methods | Symmetric | | | | Random | | | |
|---|---|---|---|---|---|---|---|---|---|
| | | 0.2 | 0.4 | 0.6 | 0.8 | 0.2 | 0.4 | 0.6 | 0.8 |
| CIFAR-10 | FedAvg | 76.84±0.91 | 63.34±1.82 | 43.83±0.51 | 22.13±1.25 | 76.24±1.58 | 59.19±1.01 | 46.80±2.63 | 21.80±0.28 |
| | LC | 79.14±0.35 | 63.57±0.61 | 44.33±1.13 | 22.98±1.60 | 74.96±1.92 | 61.49±3.02 | 40.52±2.18 | 23.84±3.37 |
| | FedProx | 70.54±0.57 | 59.35±0.65 | 45.61±0.97 | 22.70±1.10 | 68.51±0.92 | 58.61±0.38 | 43.97±1.06 | 24.64±2.59 |
| | Co-teaching | 78.64±0.45 | 70.60±0.47 | 48.63±0.57 | 21.06±2.10 | 75.11±0.39 | 59.00±1.19 | 31.30±2.03 | 17.10±3.78 |
| | T-revision | 69.16±6.20 | 51.86±6.64 | 31.93±2.56 | 15.27±1.87 | 64.69±5.08 | 46.22±1.17 | 31.81±2.83 | 17.12±0.73 |
| | FedDyn | 70.88±0.77 | 58.58±1.14 | 42.83±1.23 | 20.70±1.66 | 70.13±0.99 | 58.91±3.06 | 42.11±2.84 | 25.21±2.98 |
| | FedBN | 67.82±0.91 | 53.49±0.85 | 39.33±2.52 | 19.50±0.99 | 66.66±4.69 | 58.20±1.58 | 41.38±1.89 | 22.66±2.03 |
| | Scaffold | 64.02±0.13 | 55.50±0.96 | 37.48±2.16 | 15.10±0.43 | 59.13±0.83 | 50.36±1.54 | 34.73±4.12 | 18.23±1.66 |
| | FedDPCont | **84.77±0.12** | **75.75±1.96** | **55.50±1.33** | **24.64±0.55** | **82.15±0.24** | **72.69±1.57** | **59.06±1.38** | **27.55±1.49** |
| CIFAR-100 | FedAvg | 47.78±0.50 | 32.63±0.27 | 20.32±0.51 | 10.62±0.26 | 47.75±0.29 | 31.06±0.79 | 20.14±0.32 | 9.71±0.43 |
| | LC | 48.92±0.42 | 33.15±0.23 | 20.39±0.36 | 10.43±0.45 | 49.03±0.17 | 32.67±0.75 | 19.78±0.67 | 10.13±0.36 |
| | FedProx | 32.14±0.27 | 24.68±0.11 | 16.52±0.77 | 8.85±0.60 | 31.77±0.30 | 25.03±0.47 | 17.16±0.64 | 8.84±0.50 |
| | Co-teaching | 41.15±0.28 | 29.81±0.72 | 18.01±0.28 | 8.73±1.08 | 40.55±1.79 | 28.51±1.41 | 18.47±1.95 | 6.56±1.38 |
| | T-revision | 48.21±0.56 | 31.35±0.46 | 17.41±0.22 | 7.79±0.28 | 48.24±0.47 | 30.91±0.55 | 16.95±0.78 | 7.46±0.20 |
| | FedDyn | 31.73±0.79 | 23.35±0.23 | 15.53±0.21 | 7.82±0.04 | 32.22±0.35 | 23.83±0.42 | 16.27±0.59 | 7.86±0.10 |
| | FedBN | 40.71±1.19 | 25.61±0.53 | 14.52±0.18 | 6.64±0.32 | 38.96±0.86 | 24.54±0.86 | 13.52±0.73 | 6.63±0.17 |
| | Scaffold | 31.56±0.20 | 24.85±0.38 | 14.42±0.93 | 2.10±0.35 | 28.49±0.75 | 21.74±0.48 | 11.19±1.23 | 1.97±0.44 |
| | FedDPCont | **53.39±0.43** | **34.99±1.66** | **21.35±0.69** | **11.02±0.66** | **51.73±0.36** | **34.43±0.72** | **21.35±0.72** | **10.64±0.43** |

methods. We are aware that there are other noisy learning methods that achieve impressive performance, e.g., DivideMix (Li et al., 2020a). However, their underlying semi-supervised learning mechanisms and mix-up data augmentation (Zhang et al., 2018) methods introduce massive training cost and are out of the scope of this paper. We leave discussions related to the computation cost and performance comparisons with such method to Appendix D.1. The local updating iteration $E$ is 5 and the federated fraction $\lambda$ is 0.1. The architecture of the network is ResNet-18 (He et al., 2016) for CIFAR dataset and ResNet-50 (He et al., 2016) with ImageNet (Deng et al., 2009) pre-trained weight for Clothing-1M. The local learning rate $\alpha_l$ is 0.01 and the batch size is 32. The total communication round with the server $R$ is 300 and differential privacy $\epsilon$ are 3.58, 5.98 and 3.95 for CIFAR-10, CIFAR-100 and Clothing-1M, respectively to keep $e^\epsilon/(e^{\epsilon+K-1})$ 0.2 in Section 4.1. All the experiments are run for 3 times with different random seeds to validate the generality of our methods. The details of the implementation of every baseline method in the FL setting can be found in the Appendix.

## 5.2 SYNTHETIC OPEN-SET LABEL NOISE

There are two strategies:

- Symmetric: We first add symmetric label noise (Xia et al., 2019; Han et al., 2018) to dataset $D$ and get $\tilde{D}$, then distribute $\tilde{D}$ to $\tilde{D}_c, \forall c$ following the uniform allocation in Section 3.1. The transition matrix $T$ for the symmetric label noise satisfies $T_{ij} = \eta/(K-1), \forall i \neq j$ and $T_{ii} = 1 - \eta, \forall i \in [K]$, where $\eta \in \{0.2, 0.4, 0.6, 0.8\}$ is the average noise rate.

- Random: We first add random label noise (Zhu et al., 2022) to dataset $D$ and get $\tilde{D}$, then distribute $\tilde{D}$ to $\tilde{D}_c, \forall c$ following the non-uniform allocation in Section 3.1. The $T$ of random noise is generated as follows. The diagonal elements of $T$ for the random label noise is generated by $\eta + \mathsf{Unif}(-0.05, 0.05)$, where $\eta$ is the average noise rate, $\mathsf{Unif}(-0.05, 0.05)$ is the uniform distribution bounded by $-0.05$ and $0.05$. The off-diagonal elements in each row of $T$ follow the Dirichlet distribution $(1 - T_{ii}) \cdot \mathsf{Dir}(\mathbf{1})$, where $\mathbf{1} = [1, \cdots, 1]$ ($K-1$ values). The random strategy is more practical than the symmetric one.

**Results and Discussion** Table 1 shows FedDPCont is significantly better than all the baseline methods in the symmetric strategy across almost all the noise rate settings. It is also better than the other methods in most settings of the random strategy and can always be the top-2. FedDPCont is very competitive in all the settings. Table 1 also shows directly applying the methods for centralized learning with noisy labels cannot be statistically better than the traditional federated learning solution (FedAvg) and its adapted version (FedProx), indicating the openset label noise in FL is indeed challenging and special treatments are necessary to generalize the centralized solution to the FL setting. We also report the accuracy of the last epoch in Table **??** and **??** in Appendix. FedDPCont also stands out in most cases, showing its stability.

Table 2: The performance (the best accuracy) of all methods on CIFAR-N and Clothing-1M

| Datasets | CIFAR-10 | | | CIFAR-100 | Clothing-1M |
|---|---|---|---|---|---|
| Methods | Worst | Random | Aggregate | Fine | 1M Noisy Training |
| FedAvg | 46.55±7.82 | 59.69±4.88 | 66.41±6.52 | 22.65±2.29 | 70.27 |
| LC | 46.67±8.21 | 59.27±5.72 | 67.27±4.76 | 22.59±1.66 | 70.05 |
| FedProx | 58.47±0.97 | 69.35±0.62 | 74.48±1.00 | 35.33±0.35 | 65.96 |
| Co-teaching | 24.80±2.27 | 47.34±21.05 | 62.04±11.26 | 17.83±0.39 | 40.33 |
| T-revision | 57.85±19.44 | 55.06±8.40 | 63.40±9.99 | 22.18±1.44 | 66.95 |
| FedBN | 63.07±3.29 | 73.02±1.45 | 77.55±2.16 | 37.59±0.61 | - |
| FedDPCont | **63.50±5.63** | **73.68±4.35** | **81.86±1.09** | **40.60±1.91** | **70.88** |

Table 3: The influence of different $\epsilon$ on the performance.

| $\epsilon = 1$ | $\epsilon = 2$ | $\epsilon = 4$ | $\epsilon = 8$ | $\epsilon = 100$ | $\epsilon = 3.58$ |
|---|---|---|---|---|---|
| 72.47±2.64 | 71.60±1.96 | 72.27±1.87 | 73.00±1.96 | 73.75±2.38 | 72.44±1.52 |

## 5.3 REAL-WORLD LABEL NOISE

We also test the performance on two real-world datasets: CIFAR-N (Wei et al., 2022b) and Clothing-1M (Xiao et al., 2015). Different from the benchmark datasets, these datasets are corrupted naturally. Clothing-1M is collected from the real website where both data and labels are from the real users. The noisy ratio is about 0.4 in Clothing-1M. CIFAR-N consists of CIFAR-10 and CIFAR-100. $\tilde{D}_c$ is generated according to the random setting given in Section 5.2. The labels of CIFAR-N are collected from the human annotation. There are three levels of noisy ratio in CIFAR-10, *worst*, *aggregate* and *random* while there is only one noisy level in CIFAR-100. It can be found that FedDPCont outperforms all the baseline methods in the real-world dataset, showing great potential in practical usage.

## 5.4 EFFECT OF DP LEVEL

According to Section 4.1 and 4.2, label communication and peer gradient updates at local clients are two key steps in FedDPCont. $\epsilon$ is the parameter to control the level of DP protection. Following Ghazi et al. (2021), we study the influence of $\epsilon$ on the performance. We select the CIFAR-10 corrupted by random noise whose ratio is 0.4. All the experiments are run with 10 random seeds. In terms of the randomness of model initialization and the noise generation, it can be found that FedDPCont is stable with the change of $\epsilon$, which agrees with our theoretical guarantee.

## 6 CONCLUSION

We have defined openset label noise in FL and proposed FedDPCont to use globally communicated contrastive labels to prevent local models from memorizing openset noise patterns. We have proved that FedDPCont is able to approximate a centralized solution with strong theoretical guarantees. Our experiments also verified the advantage of FedDPCont. Admittedly, FedDPCont is only tested with different label noise regimes with synthetic data partitions. Future works include testing FedDPCont with real-world FL data partitions and real-world clients such as mobile devices.

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

**Roadmap** The appendix is composed as follows. Section A presents all the notations and their meaning we use in this paper. Section C introduces the implementation details of the experiments and how to apply the centralized training methods to FL. Section D shows the experiment results with more details that are not given in the main paper due to the page limit.

## A  NOTATION TABLE

Table 4: Table of notations used in the paper

| Notation | Explanation |
|---|---|
| $\eta$ | Noisy ratio |
| $C$ | Total number of clients |
| $c$ | Client $c$ in federated learning |
| $\tilde{Y}$ | Random variables for the noisy label |
| $\hat{Y}$ | Random variables for the output of the model |
| $K$ | Number of classes in $\mathcal{Y}$ |
| $T$ | Transition matrix |
| $\mathbb{P}$ | The probability |
| $\mathbb{E}$ | The expectation |
| $\lambda$ | Federated fraction to control the number of clients in every round |
| $\alpha_g$ | The global step size on the server side |
| $\alpha_l$ | The local learning rate on the client side |
| $L, \ell$ | The loss, the loss function |
| $R$ | The global communication round |
| $E$ | The local updating round |
| $T_{\text{DP}}$ | Differential privacy transition matrix |
| $T$ | Transition matrix |
| $\mathcal{A}$ | The label communication algorithm |
| $\check{Y}_c$ | Labels protected by differential privacy |
| $\tilde{\boldsymbol{p}}_n^c$ | one-hot encoding of $\tilde{y}_n^c$ |
| $\check{\boldsymbol{p}}$ | The posterior label distribution after differential privacy corruption |
| $\theta, \theta_c^r$ | The model parameters, the model parameters of client $c$ at $r$-th round |
| $e_1, e_2$ | The noisy ratio of class 1 and 2 of the global dataset in binary classification |
| $e_1^k, e_2^k$ | The noisy ratio of class 1 and 2 of the local client $k$ in binary classification |
| $m_1$ | The number of samples which are wrongly labeled from 1 to 2 in binary classification |
| $m_2$ | The number of samples which are wrongly labeled from 2 to 1 in binary classification |
| $\Delta_c^{(r)}$ | The variation of model parameters in $r$-th round of the client $c$ |
| $X, Y$ | Random variables for the feature and label |
| $\mathcal{X}, \mathcal{Y}$ | The space of $X, Y$ |
| $\boldsymbol{f}_c, \boldsymbol{f}_g$ | The client model, The global model |
| $N, N_c$ | Total number of samples, number of samples in client $c$ |
| $(x_n^c, y_n^c)$ | The $n$-th example in the client $c$ |
| $D_c := \{(x_n^c, y_n^c)\}_{n \in [N_c]}$ | Dataset of client $c$ |
| $D := \{(x_n, y_n)\}_{n \in [N]}$ | Dataset |
| $I_k := \{c \mid \mathbb{1}_{c,k} = 1, c \in [C]\}$ | The vector indicating whether client $c$ can access class $k$ or not |

## B  PROOFS AND ANALYSES

In this section, we present all the proofs of the theorems.

### B.1  PROOF OF THEOREM 1

*Proof.* Denote by $\mathcal{A}$ the label communication algorithm, where the input is $y$ and the output is $y_{\text{DP}}$. Then after flipping the label $y$ according to the noise transition matrix $T$, we have

$$\mathbb{P}(\mathcal{A}(y) = y_{\text{DP}}) = \begin{cases} \frac{e^\epsilon}{e^\epsilon + K - 1}, & \text{if } y_{\text{DP}} = y, \\ \frac{1}{e^\epsilon + K - 1}, & \text{if } y_{\text{DP}} \neq y. \end{cases}$$

Accordingly, for another label $y'$, we have

$$\mathbb{P}(\mathcal{A}(y') = y_{\text{DP}}) = \begin{cases} \frac{e^\epsilon}{e^\epsilon + K - 1}, & \text{if } y_{\text{DP}} = y', \\ \frac{1}{e^\epsilon + K - 1}, & \text{if } y_{\text{DP}} \neq y'. \end{cases}$$

Then the quotient of two probabilities can be upper bounded by

$$\frac{\mathbb{P}(\mathcal{A}(y) = y_{\text{DP}})}{\mathbb{P}(\mathcal{A}(y') = y_{\text{DP}})} \leq e^\epsilon.$$

With Definition 2, we know the above equation is exactly the definition of $\epsilon$-labelDP, i.e., the label communication algorithm is $\epsilon$-labelDP.

$\square$

### B.2 Proof of Theorem 2

*Proof.* The centralized peer loss on $\mathcal{D}$ is

$$\mathbb{E}_{\mathcal{D}}[\ell_{\text{peer}}(f(X), \widetilde{Y})] = \mathbb{E}_{\mathcal{D}}\left[\ell(f(X), \widetilde{Y}) - \beta \cdot \mathbb{E}_{\mathcal{D}_{\widetilde{Y}'|\widetilde{D}}}[\ell(f(X), \widetilde{Y}')]\right],$$

where $\tilde{Y}'$ is the random variable whose distribution is the noisy label distribution. For each client $c$, the local FedDPCont loss is

$$\mathbb{E}_{\mathcal{D}_c}[\ell_{\text{FedDPCont}}(f(X_c), \widetilde{Y}_c)] = \mathbb{E}_{\mathcal{D}_c}\left[\ell(f(X_c), \widetilde{Y}_c) - \beta \cdot \mathbb{E}_{\mathcal{D}_{\widetilde{Y}'|\widetilde{D}}}[\ell(f(X_c), \widetilde{Y}')]\right],$$

.

Denote by $\mathbb{P}(\mathcal{D}_c|\mathcal{D})$ the probability of drawing a data point from client $c$. We have

$$\sum_{c \in [C]} \mathbb{P}(\mathcal{D}_c|\mathcal{D}) = 1.$$

Then

$$\sum_{c \in [C]} \mathbb{P}(\mathcal{D}_c|\mathcal{D})\mathbb{E}_{\mathcal{D}_c}[\ell_{\text{FedDPCont}}(f(X_c), \widetilde{Y}_c)]$$

$$= \sum_{c \in [C]} \mathbb{P}(\mathcal{D}_c|\mathcal{D})\mathbb{E}_{\mathcal{D}_c}\left[\ell(f(X_c), \widetilde{Y}_c) - \beta \cdot \mathbb{E}_{\mathcal{D}_{\widetilde{Y}'|\widetilde{D}}}[\ell(f(X_c), \widetilde{Y}')]\right]$$

$$= \mathbb{E}_{\mathcal{D}}\left[\ell(f(X), \widetilde{Y}) - \beta \cdot \mathbb{E}_{\mathcal{D}_{\widetilde{Y}'|\widetilde{D}}}[\ell(f(X), \widetilde{Y}')]\right]$$

$$= \mathbb{E}_{\mathcal{D}}[\ell_{\text{peer}}(f(X), \widetilde{Y})].$$

Each round may include multiple epochs. Suppose there are $t$ local epochs. The variation of model parameters in the $r$-th round of the local training in client $c$ can be decomposed by

$$\Delta_c^{(r)} := \theta_c^{(r+1)} - \theta_c^{(r)}$$

$$= \theta_c^{(r+1,t)} - \theta_c^{(r+1,t-1)} + \theta_c^{(r+1,t-1)} + \cdots - \theta_c^{(r,1)}$$

$$= \frac{\partial \mathbb{E}_{\mathcal{D}_c}[\ell_{\text{FedDPCont}}(f(X_c), \widetilde{Y}_c; \theta_c)]}{\partial \theta_c}\bigg|_{\theta = \theta_c^{(r+1,t-1)}} + \cdots + \frac{\partial \mathbb{E}_{\mathcal{D}_c}[\ell_{\text{FedDPCont}}(f(X_c), \widetilde{Y}_c; \theta_c)]}{\partial \theta_c}\bigg|_{\theta = \theta_c^{(r+1,1)}}.$$

Therefore,

$$\sum_{c \in [C]} \mathbb{P}(\mathcal{D}_c|\mathcal{D})\Delta_c^{(r)} = \frac{\partial \sum_{c \in [C]} \mathbb{P}(\mathcal{D}_c|\mathcal{D})\mathbb{E}_{\mathcal{D}_c}[\ell_{\text{FedDPCont}}(f(X_c), \widetilde{Y}_c; \theta_c^{(r+1,t-1)})]}{\partial \theta_c}$$

$$+ \cdots + \frac{\partial \sum_{c \in [C]} \mathbb{P}(\mathcal{D}_c|\mathcal{D})\mathbb{E}_{\mathcal{D}_c}[\ell_{\text{FedDPCont}}(f(X_c), \widetilde{Y}_c; \theta_c^{(r+1,1)})]}{\partial \theta_c}$$

$$= \frac{\partial \mathbb{E}_{\mathcal{D}}[\ell_{\text{peer}}(f(X), \widetilde{Y}; \theta^{(r+1,t-1)})]}{\partial \theta}$$

$$+ \cdots + \frac{\partial \mathbb{E}_{\mathcal{D}}[\ell_{\text{peer}}(f(X), \widetilde{Y}; \theta^{(r+1,1)})]}{\partial \theta}$$

$$= \Delta^{(r)}.$$

$\square$

### B.3 DETAILS ABOUT THE NECESSITY OF USING A GLOBAL PRIVATE LABEL

To be more concrete, in Liu & Guo (2020), for each example $(x_n, \tilde{y}_n)$, peer loss defines as (an equivalent form):

$$\ell_{\text{PL}}(\boldsymbol{f}(x_n), \tilde{y}_n) := \ell(\boldsymbol{f}(x_n), \tilde{y}_n) - \ell(\boldsymbol{f}(x_n), \tilde{y}_{n'}), \tag{4}$$

where $\tilde{y}_{n'}$ is a randomly sampled peer label. Later as a follow-up work (Cheng et al., 2020), $\ell_{\text{CORES}}$ was proposed as a more stable version of $\ell_{\text{PL}}$ which has the same expectation as $\ell_{\text{PL}}$:

$$\ell_{\text{CORES}}(\boldsymbol{f}(x_n), \tilde{y}_n) = \ell(\boldsymbol{f}(x_n), \tilde{y}_n) - \mathbb{E}_{\mathcal{D}_{\tilde{Y}|\tilde{D}}}[\ell(\boldsymbol{f}(x_n), \tilde{Y}], \tag{5}$$

where $\mathcal{D}_{\tilde{Y}|\tilde{D}}$ is the distribution of $\tilde{Y}$ given dataset $\tilde{D}$. Peer loss and $\ell_{\text{CORES}}$ have strong consistency guarantees. Consider a binary classification problem and let $e_1 := \mathbb{P}(\tilde{Y} = 2|Y = 1)$ and $e_2 = \mathbb{P}(\tilde{Y} = 1|Y = 2)$. Then it was proved in Liu & Guo (2020) the following robustness of peer loss:

**Proposition 3** (Robustness of peer loss (Liu & Guo, 2020)). *Peer loss is invariant to label noise:*

$$\mathbb{E}_{\tilde{\mathcal{D}}}[\ell_{PL}(f(X), \tilde{Y})] = (1 - e_1 - e_2) \cdot \mathbb{E}_{\mathcal{D}}[\ell_{PL}(f(X), Y)].$$

*Moreover, when $\mathbb{P}(Y = 1) = 0.5$ and $\ell$ is the 0-1 loss, minimizing peer loss on noisy distribution $\tilde{\mathcal{D}}$ is equivalent to minimizing 0-1 loss on clean distribution $\mathcal{D}$.*

Can we then follow the above idea and implement either $\ell_{\text{PL}}$ or $\ell_{\text{CORES}}$ by requiring each client to sample the "peer label" $\tilde{y}_{n'}$ locally? Unfortunately, the answer is no. There are two **technical challenges**:

*First, sampling peer labels locally leads to __wrong__ results.* A local sampling for the private label will lead to a distribution that does not capture the global one on $\mathbb{P}(\tilde{Y})$, then challenge the theoretical guarantees of the existing results. To see this, we consider a binary classification problem. Assume that we have two clients $c = 1$ and $c = 2$, where client 1 can only access noisy labels 1 and client 2 only accesses noisy labels 2, respectively. Suppose the number of data points in each class (globally) is $N_1 = N_2 = N/2$. If there are $m_1$ samples that are wrongly labeled from $Y = 1$ to $\tilde{Y} = 2$ and $m_2$ samples that are wrongly labeled from $Y = 2$ to $\tilde{Y} = 1$, respectively, we can know the global noisy ratios are $e_1 = \mathbb{P}(\tilde{Y} = 2|Y = 1) = 2m_1/N$ and $e_2 = \mathbb{P}(\tilde{Y} = 1|Y = 2) = 2m_2/N$, respectively. For centralized training, we know from Proposition 3 that there is an invariant property. However, due to the openset, the locally noisy ratio differs from the globally noisy ratio and the invariant property is **broken**. Specifically, the local noisy ratios are $e_1^1 = \mathbb{P}(\tilde{Y}_1 = 2|Y_1 = 1) = 0$ and $e_2^1 = \mathbb{P}(\tilde{Y}_1 = 1|Y_1 = 2) = 1$ where $Y_1$ is the label and $\tilde{Y}_1$ is the corrupted label in client 1. Then the invariant property in Proposition 3 becomes

$$\mathbb{E}_{\tilde{\mathcal{D}}_1}[\ell_{\text{PL}}(f(X), \tilde{Y})] = (1 - e_1^1 - e_2^1) = 0,$$

which is a constant for any model $f$. Therefore, peer labels need to be redesigned in FL with openset noisy labels.

*Second, there are privacy concerns in redesigning peer labels.* Intuitively, since we know local sampling fails, the global information is inevitable in redesigning peer labels. Therefore, the privacy issues need to be addressed in label communications.

## C   IMPLEMENTATION DETAILS

**Platform and Programming Environment**   We train our model on NVIDIA RTX A5000 server with torch and torchvision 1.10 and 0.11, respectively. The details of the baseline methods are as follows.

**Loss correction**   We apply FedAvg in the first 150 rounds to make the weight stable. At the 150th round, the transition matrix of every client will be estimated according to the confidential score of 95%. The predicted label whose confidential score is over 95% is considered as the ground truth so that we can get every transition matrix of every client. We apply loss correction in the rest 150 rounds according to Equation 1.

Table 5: Comparison of DivideMix and FedDPCont in terms of time and number of epochs on benchmark dataset. R stands for random noise in Section 5.2. Compared with DivideMix, FedDPContis lightweight and can produce relatively reliable results.

| Dataset | Methods | 0.2 | 0.4 | 0.6 | 0.8 | Epochs | Time (hr) |
|---|---|---|---|---|---|---|---|
| CIFAR-10 | DivideMix | 79.28±0.33 | 65.62±4.48 | 53.28±2.16 | 20.70±5.00 | 300 | 22.33±1.15 |
| | FedDPCont | **84.77±0.11** | **75.75±1.96** | **55.50±1.33** | **24.64±0.55** | 300 | 4.17±0.29 |
| CIFAR-10 (R) | DivideMix | 70.31±2.19 | 59.24±1.90 | 48.91±4.59 | 24.29±2.19 | 300 | 27.67±1.03 |
| | FedDPCont | **82.15±0.24** | **72.69±1.57** | **59.06±1.38** | **27.55±1.49** | 300 | 3.90±0.17 |
| CIFAR-100 | DivideMix | **58.31±0.47** | **46.62±0.37** | **31.9±0.85** | **19.87±0.52** | 300 | 30.33±2.52 |
| | FedDPCont | 53.39±0.43 | 34.99±1.66 | 21.35±0.69 | 11.02±0.66 | 300 | 3.87±0.13 |
| CIFAR-100 (R) | DivideMix | **57.31±0.08** | **45.6±0.51** | **30.95±0.57** | **19.40±0.53** | 300 | 29.33±1.53 |
| | FedDPCont | 51.73±0.36 | 34.43±0.72 | 21.35±0.13 | 10.64±0.43 | 300 | 4.28±0.25 |

**Co-teaching**   Co-teaching uses two same networks to distinguish the noisy data and the clean data. Similarly, we initialize two same networks when the client initializes and update the two clients in the same way as the original co-teaching network. The server also keeps two models. In every communication round, the weights of the two models will average correspondingly.

**T-revision**   T-revision consists of three steps: estimation of $T$, loss correction, and T-revision. In the first 20 communication rounds, the selected clients update the weight at every communication round and all the clients estimate $T_c$. After the 20th round, the selected clients at every communication apply forward loss correction for another 140 rounds. After the 160th round, we apply T-revision.

**DivideMix**   DivideMix uses two same networks to distinguish the noisy label. One network is used to assign the pseudo label, the other network is used to the classification. The pseudo label is generated by a Gaussian mixture process. In addition, DivideMix uses mix-up data augmentation to boost performance. In FL paradigm, every client will maintain two clients and do the same operation as the centralized training in DivideMix.

**Other Baseline Methods**   For the other baseline methods, we follow the original settings in their papers.

# D  EXPERIMENT RESULTS

## D.1  DIVIDEMIX DETAILS

Compared with DivideMix, FedDPCont is a lightweight method. Due to the mix-up augmentation method and dual-model architecture design, DivideMix needs more time to converge. We compare the performance of DivideMix and FedDPCont in terms of the epoch and the training time. All the experiments are done on a server with an AMD EPYC 7513 32-Core Processor and RTX A5000 NVIDIA GPU to guarantee the training time is calculated fairly. The results on benchmark and real-world datasets are given in Table 5 and 6.

We can find that FedDPCont needs much less time than DivideMix in all cases and outperforms on all CIFAR-10 datasets for both benchmark and real-world cases. FedDPCont depends heavily on the estimation of the distribution of the dataset from label communication as given in Section 4.1. When the data belonging to each class is fewer or the noisy ratio is higher, the difficulty of precise estimation becomes much larger. Compared with FedDPCont, DivideMix uses another model to generate the pseudo label so that the performance can be less sensitive to the heterogeneity but will be much slower. In the practical usage, FedDPCont is a reliable choice in terms of speed and performance.

Table 6: Comparison of DivideMix and FedDPCont in terms of time and number of epochs on the noisy real-world dataset. Compared with DivideMix, FedDPContis lightweight and can produce relatively reliable results.

| Dataset | Methods | Accuracy | Epochs | Time (hr) |
|---|---|---|---|---|
| CIFAR-10-N-Worst | DivideMix | 59.50±5.90 | 300 | 33.00±1.73 |
| | FedDPCont | **63.50±5.63** | 300 | 3.76±0.31 |
| CIFAR-10-N-Random | DivideMix | 66.45±2.69 | 300 | 21.50±0.71 |
| | FedDPCont | **73.68±4.35** | 300 | 3.43±0.12 |
| CIFAR-10-N-Aggregate | DivideMix | 71.98±2.27 | 300 | 25.50±0.71 |
| | FedDPCont | **81.86±1.09** | 300 | 3.83±0.29 |
| CIFAR-100-N | DivideMix | **45.66±0.15** | 300 | 13.85±0.21 |
| | FedDPCont | 40.60±1.91 | 300 | 2.63±0.18 |

