# OpenReview forum: "Federated Learning with Local Openset Noisy Labels"
_ICLR.cc/2024/Conference — Submitted to ICLR 2024_

### Official Review · Reviewer_L6ru · 2023-10-28

**Soundness:** 3 good
**Presentation:** 3 good
**Contribution:** 3 good
**Rating:** 5
**Confidence:** 4

**Summary:**

This paper tries to mitigate the open-set noisy labels in a federated setting based on the loss correction method in centralized noisy label learning. In a centralized setting, the loss correction method needs to estimate a label transformation matrix to correct the overall loss values, which can be unachievable for federated learning due to the fact the local noise pattern is different overall clients in “openset federated noisy label” setting. To tackle this problem, this paper proposes a method of sharing local flipped labels to servers with a privacy guarantee from label differential privacy (LDP). Then, the server can calculate the posterior label distribution and send it to a client. On local training, each client utilizes the posterior label distribution and LDP distribution to fix the local training loss function. Also, contrastive learning is utilized in local training to boost performance.

**Strengths:**

- Originality: The motivation of this paper is meaningful and reasonable. That is, we need to address the construction of a transformation matrix for label noise in federated learning without sharing direct local label distributions.

- Quality: A detailed explanation of how the idea was developed and the details of each step in the algorithm are explained. The datasets used in experiments are plentiful.

**Weaknesses:**

- Experiment:
  - How is data partitioned in an experiment? In federated learning, there are IID and non-IID partitions (the current discussion is based on ground truth labels/features), while in the paper it is not mentioned.
  - the centralized noisy label learning baselines used in this paper are LC, Co-teaching, T-revision, which were proposed several years ago. Could you please use some new baselines?
  - Also, authors try to avoid comparison with DivideMix by claiming that DivideMix consumes too much local computation. One suggestion is authors can use a single model version of DivideMix (mentioned as DivideMix without co-training in the original paper).
  - Authors can use [1] [ELR](https://github.com/shengliu66/ELR) as a baseline since it consumes less time compared with DivideMix, but has similar performance.
 - Another suggestion is authors can try to experiment with sharing direct label distribution information (say, if it is allowed to share privacy label information to server), and use this performance as "upper-bound" for LDP used in this paper.
 - Please use at least one latest FL noisy label method as the baseline, for example [3] [FedCorr](https://github.com/Xu-Jingyi/FedCorr) [4] [RHFL](https://github.com/FangXiuwen/Robust_FL)
- openest noisy label in federated learning: actually, the definition of "openest federated noisy label" setting is almost the same as one of the settings proposed in [2] [FedNoisy](https://arxiv.org/abs/2306.11650), though they may use different noise generation implementation as in authors' work.

[1] Liu, Sheng, Jonathan Niles-Weed, Narges Razavian, and Carlos Fernandez-Granda. "Early-learning regularization prevents memorization of noisy labels." Advances in neural information processing systems 33 (2020): 20331-20342.
[2] Liang, Siqi, Jintao Huang, Dun Zeng, Junyuan Hong, Jiayu Zhou, and Zenglin Xu. "FedNoisy: Federated Noisy Label Learning Benchmark." *arXiv preprint arXiv:2306.11650* (2023).
[3] Xu, Jingyi, Zihan Chen, Tony QS Quek, and Kai Fong Ernest Chong. "Fedcorr: Multi-stage federated learning for label noise correction." In Proceedings of the IEEE/CVF Conference on Computer Vision and Pattern Recognition, pp. 10184-10193. 2022.
[4] Fang, Xiuwen, and Mang Ye. "Robust federated learning with noisy and heterogeneous clients." Proceedings of the IEEE/CVF Conference on Computer Vision and Pattern Recognition. 2022.

**Questions:**

- How the time for DivideMix is measured? If the time is N clients train in sequence, then the result will enlarge the time difference between that of FedDPCont

---

### Official Review · Reviewer_CSq3 · 2023-10-31

**Soundness:** 2 fair
**Presentation:** 1 poor
**Contribution:** 2 fair
**Rating:** 3
**Confidence:** 3

**Summary:**

Authors propose an algorithm that would be functional under FL settings wherein local training labels are non-overlapping and possibly noisy. The algorithm uses a variant based on loss correction using something they term as "contrastive labels" that is generated by the server by aggregating noisy statistics from clients based on loss correction/transition matrix estimated by the server. They argue their method has both DP guarantees and theoretical effectiveness otherwise.

**Strengths:**

- Overall the FL setting considered by the author is quite relevant and realistic, such as
	- data centers having different data distribution
	- devices having different noise levels;

- The overall effect of the algorithm in terms of performance is quite convincing.

**Weaknesses:**

- Looking at the results in Table 3 and the limited effect of epsilon, it brings up the question: What is the actual effect of using the contrastive label
- Overall it is quite hard to follow the paper with the confusing notation:
	- usage of \hat, \tilde, \check, and \prime gets very unnecessarily confusing.
	- on page 4 authors write $-l(f(x_{n\prime}), \check{y}_n)$ , but in eqn 3 they suddenly switch to $-l(f(x_{n}), \check{y}_{n\prime})$
	- on page 5 section 4.2 what is $P(\tilde{Y}\vert\tilde{D})$
- I'm not quite sure what this means: "Intuitively, the “new” label has to be sampled globally; otherwise, the global information is missing and the negative effect of local openset label noise would induce performance degradation." There are no experiments to confirm or refute this claim. In recent papers [1] it's been observed that the addition of noise (e.g., data augmentation) can in fact improve results on heterogeneous data
- In section 5.1 authors mention that "mixup data augmentation methods introduce massive training cost". that does not seem to be the case as per the paper [2]. can the authors discuss this in more detail.

1. Azam, Sheikh Shams, et al. "Importance of Smoothness Induced by Optimizers in FL4ASR: Towards Understanding Federated Learning for End-to-End ASR." _arXiv preprint arXiv:2309.13102_ (2023).
2. Zhang, Hongyi, et al. "mixup: Beyond empirical risk minimization." _arXiv preprint arXiv:1710.09412_ (2017).

**Questions:**

- it is unclear what they mean by noise rates in Section 1 paragraph 1: as in different SNR or different distributions of noise altogether 🔗
- Based on the authors' presentation, is it right to infer that the proposed method provides a differentially private way of estimating the noise transition matrix followed by FedAvg using the transition matrix? 🔗
- on page 2: when authors cite for statement "existing popular noisy learning solutions such as loss correction", the latest paper is as old as 2017. does this mean there are no newer works that have tried to tackle this? 🔗
- on page 3, the authors say "each client c sends its model parameter", do they mean sending the gradients? this is an important consideration especially for additive noise DP since it requires clipping, wherein clipping model parameters and model updates are entirely different because of their associated norms 🔗
- It is not clear what do the authors mean by the sentence "In practice if all the elements in ... will be re-generated until client c is an openset client." 🔗
- In the example on Page 4. it is not clear how do the authors renormalize T_real to T_OptEst? 🔗

---

### Official Review · Reviewer_zvrT · 2023-10-31

**Soundness:** 3 good
**Presentation:** 2 fair
**Contribution:** 2 fair
**Rating:** 5
**Confidence:** 3

**Summary:**

The paper addresses an important and practical problem in the context of federated learning (FL) with heterogeneous data sources. It introduces the challenge of handling noisy training labels in a federated setting, where clients have limited access to label spaces. The proposed label communication mechanism, which leverages contrastive labels and label differential privacy, presents a promising approach to mitigate the impact of noisy labels.

**Strengths:**

1. The proposed label communication mechanism using "contrastive labels" and label differential privacy is innovative and appears well-suited to address the challenge of noisy labels in a federated context. The theoretical guarantees of both privacy and effectiveness add strength to the approach.

2. The paper focuses on an important problem in federated learning: dealing with noisy labels in a setting where clients have limited access to label spaces. This problem is significant and highly relevant to real-world applications.

3. The paper is well-organized and effectively communicates its contributions and methodology, making it easy to capture the main idea.

**Weaknesses:**

1. According to the standard definition of 'openset' in machine learning, the term typically refers to a scenario where there are two distinct label sets: known labels and unknown labels. In this paper, such a clear distinction between known and unknown label sets is not provided. Instead, the paper focuses on addressing a different issue related to class imbalance, which is more characteristic of non-IID federated learning. Consequently, it is not entirely accurate to claim that this paper solves the 'openset noise' problem in federated learning. Furthermore, it remains unclear how the proposed methods would perform in addressing a genuine openset noise problem, as the server might struggle to initialize T_DP without knowledge of both known and unknown labels.

2. The stability of the results in Table 3, particularly when the differential privacy noise level (e) is set to 100, raises questions. It is important to provide a detailed explanation of why the performance remains largely unaffected at this noise level and how this stability aligns with the theoretical guarantees of the proposed approach.

3. A notable weakness in this work is the absence of a comparative analysis with existing studies that aim to address label noise in federated learning, such as FedCorr (CVPR 2022 [1]), FedRN (CIKM 2022 [2]), FedNoRo (IJCAI 2023 [3]).

[1] Xu, Jingyi, et al. "Fedcorr: Multi-stage federated learning for label noise correction." Proceedings of the IEEE/CVF Conference on Computer Vision and Pattern Recognition. 2022.

[2] Kim, Sangmook, et al. "FedRN: Exploiting k-Reliable Neighbors Towards Robust Federated Learning." Proceedings of the 31st ACM International Conference on Information & Knowledge Management. 2022.

[3] Wu, Nannan, et al. "FedNoRo: Towards Noise-Robust Federated Learning by Addressing Class Imbalance and Label Noise Heterogeneity." arXiv preprint arXiv:2305.05230 (2023).

**Questions:**

Please see weaknesses

---

### Official Review · Reviewer_pQNK · 2023-11-05

**Soundness:** 3 good
**Presentation:** 2 fair
**Contribution:** 2 fair
**Rating:** 5
**Confidence:** 3

**Summary:**

This paper designs a label communication mechanism to share randomly selected labels among clients. This work preserves the privacy of shared labels by using label differential privacy and shows its performance on benchmark datasets and real-world datasets.

**Strengths:**

The issues discussed in this work have certain practical value. The proposed solution is a straightforward method to solve the noisy label problem in FL scenes.

**Weaknesses:**

Some of the unclear descriptions in the paper are confusing. Although the author uses DP to share labels since the server knows T_DP and the server can obtain the user's label distribution p_c, the risk of privacy leakage still seems to exist.

**Questions:**

1.	In related work, the author introduced the work of Yang et al., 2022; Xu et al., 2022 attempts to solve the noisy label problem in FL. Even though the patterns in which these works set noisy labels are the same across users, they still serve as the baseline for comparison in this paper. The authors should compare the proposed method with these baselines.

2.	What does the subscript $i$ in the formula below Theorem 1 stand for? What does the Table ?? in the Results and Discussion Section stand for?

3.	In Table 1, the performance of FedDPCont under the Random setting when noise rate=0.8 or 0.6 is better than that under the Symmetric setting. What is the reason or mechanism behind this?

4.	If the server does not aggregate the DP private label feedback by users, but only uses the DP private label of any individual user to perform the steps in line 9 of Algorithm 1, can the server calculate the real local label distribution of the user? Does this imply a leak of privacy?

---

### Meta-Review · Area_Chair_PJz8 · 2023-12-04

**Metareview:**

The paper proposes a label DP method for addressing the openset noisy labels problem in federated learning.

Strengths: the reviewers find the research question interesting and relevant.

Weaknesses: many reviewers find the lack of comparisons with related methods unacceptable. The reviewers also raise a number of other questions about the method, but there is no response from the authors.

**Justification For Why Not Higher Score:**

All reviewers recommend rejection. Many comment on insufficient comparisons with related work, which does seem a serious issue.

**Justification For Why Not Lower Score:**

N/A

---

### Decision · Program_Chairs · 2024-01-16

Reject